# Sex-Specific Effects of Early-Life Iron Deficiency and Prenatal Choline Treatment on Adult Rat Hippocampal Transcriptome

**DOI:** 10.3390/nu15061316

**Published:** 2023-03-07

**Authors:** Shirelle X. Liu, Tenille K. Fredrickson, Natalia Calixto Mancipe, Michael K. Georgieff, Phu V. Tran

**Affiliations:** 1Department of Pediatrics, University of Minnesota, Minneapolis, MN 55455, USA; 2Research Informatic Solutions, Minnesota Supercomputing Institute, University of Minnesota, Minneapolis, MN 55455, USA

**Keywords:** iron deficiency, choline, sex difference, hippocampus, transcriptome

## Abstract

Background: Fetal-neonatal iron deficiency (ID) causes long-term neurocognitive and affective dysfunctions. Clinical and preclinical studies have shown that early-life ID produces sex-specific effects. However, little is known about the molecular mechanisms underlying these early-life ID-induced sex-specific effects on neural gene regulation. Objective: To illustrate sex-specific transcriptome alterations in adult rat hippocampus induced by fetal-neonatal ID and prenatal choline treatment. Methods: Pregnant rats were fed an iron-deficient (4 mg/kg Fe) or iron-sufficient (200 mg/kg Fe) diet from gestational day (G) 2 to postnatal day (P) 7 with or without choline supplementation (5 g/kg choline) from G11–18. Hippocampi were collected from P65 offspring of both sexes and analyzed for changes in gene expression. Results: Both early-life ID and choline treatment induced transcriptional changes in adult female and male rat hippocampi. Both sexes showed ID-induced alterations in gene networks leading to enhanced neuroinflammation. In females, ID-induced changes indicated enhanced activity of oxidative phosphorylation and fatty acid metabolism, which were contrary to the ID effects in males. Prenatal choline supplementation induced the most robust changes in gene expression, particularly in iron-deficient animals where it partially rescued ID-induced dysregulation. Choline supplementation also altered hippocampal transcriptome in iron-sufficient rats with indications for both beneficial and adverse effects. Conclusions: This study provided unbiased global assessments of gene expression regulated by iron and choline in a sex-specific manner, with greater effects in female than male rats. Our new findings highlight potential sex-specific gene networks regulated by iron and choline for further investigation.

## 1. Introduction

Iron deficiency (ID) during the fetal-neonatal period causes long-term neurocognitive and emotional abnormalities [1,2,3,4]. In clinical populations, there is evidence that supports sex-specific outcomes of early-life ID and its treatments, including differential effects on fecal microbiota [5], brain development [6], verbal fluency [7], and risks of cognitive disorders such as schizophrenia [2]. Sex-specific physiologic and behavioral effects are also observed in preclinical models [8,9,10,11]. Early postnatal ID anemia causes sex-specific neuroinflammatory response [8], seizure vulnerability [9], social and affective behavioral abnormalities [10], and vasculature dysfunction [11]. However, analysis of sex-specific changes in gene regulation remains limited, including hippocampal transcriptomic changes from our phlebotomy-induced ID anemia (PIA) mouse model [8] and placental transcriptomic changes from a maternal ID mouse model [12]. Thus, additional unbiased global transcriptomic studies are needed to pinpoint the sex-specific gene networks altered by an early-life ID that can be further investigated for their contribution to the sex-specific long-term neurobehavioral effects and disease vulnerabilities [13].

The hippocampus is an important brain structure that mediates learning, memory, and cognitive functions, which can be influenced by sex [14,15]. The effects of early-life ID on the development and function of the rat hippocampus have been well documented [16,17,18,19,20,21]. However, more studies are needed to precisely determine the long-term sex-specific changes in hippocampal gene regulation in order to provide a comprehensive understanding of the negative and long-lasting effects of early-life ID, as well as to develop more effective, individualized treatment strategies.

Transcriptomic analysis using Next-Generation Sequencing (NGS) technology offers a unique unbiased approach to identifying global changes in the transcriptome. NGS findings can be annotated into changes in relevant biological processes, canonical pathways, and disease states utilizing knowledge-based databases such as Ingenuity Pathway Analysis (IPA, Qiagen Inc.), Molecular Signatures Database (MSigDB), and Gene Ontology (GO) [22,23,24]. The present study analyzed and compared long-term changes in hippocampal gene regulation between adult male and female rats that were iron-deficient or iron-sufficient during the fetal-neonatal period and between those with and without prenatal choline treatment.

## 2. Materials and Methods

### 2.1. Animals

Gestational day (G) two pregnant Sprague–Dawley rats were purchased from Charles River Laboratories (Wilmington, MA, USA). Rats were maintained in a 12-h:12-h light/dark cycle with ad-lib food and water. ID was induced by dietary manipulation as described previously [21]. In brief, to generate iron-deficient pups, pregnant dams were given a purified iron-deficient diet (4 mg Fe/kg, TD.80396; Harlan Teklad) from G2 to postnatal day (P) 7 when lactating dams were switched to a purified iron-sufficient diet (200 mg Fe/kg, TD.09256; Harlan Teklad). Control iron-sufficient pups were generated from pregnant dams maintained on the iron-sufficient diet. Both diets were the same in all respects except for the iron (ferric citrate) content. Half of the dams on the iron-sufficient or iron-deficient diet received dietary choline supplementation (5.0 g/kg choline chloride supplemented; iron-sufficient with choline: TD.1448261, iron-deficient with choline: TD.110139; Harlan Teklad) from G11–G18, while the other half of the dams received their iron-sufficient or iron-deficient diet with standard choline content (1.1 g/kg). Thus, dams and their litters were randomly assigned to one of the four groups based on the maternal diet: iron-deficient with choline supplementation (IDch), iron-deficient without supplemental choline (ID), always iron-sufficient with choline supplementation (ISch), always iron-sufficient without supplemental choline (IS). Details of the diet contents have been described in our previous study [25]. All litters were culled into eight pups with an equal number of each sex at birth. The rats were weaned at P21. Littermates were separated by sex and housed in groups of two. To avoid litter-specific effects, one rat per litter was used in the RNAseq experiments. The University of Minnesota Institutional Animal Care and Use Committee approved all experiments in this study (Protocol # 2001-37802A).

### 2.2. Hippocampal Dissection

Rats were euthanized on P65 by an injection of pentobarbital (100 mg/kg, intraperitoneal). Brains were removed and bisected along the midline on an ice-cold metal block. Hippocampus was dissected and immediately flash-frozen in liquid nitrogen and stored at −80 °C for further use.

### 2.3. RNA Isolation and Sequencing

Total RNA was isolated from the hippocampus of P65 females of all four groups (*n* = 4/group) using the RNeasy Midi Kit (Qiagen, Redwood City, CA, USA). RNA samples were submitted to the University of Minnesota Genomics Center for library preparation and sequencing. RNA quantity and quality were assessed using the RiboGreen RNA Assay kit (Invitrogen, Waltham, MA, USA) and capillary electrophoresis (Agilent BioAnalyzer 2100, Agilent, Santa Clara, CA, USA), respectively. RNA samples with a RIN score > 8.0 were used for library preparation. Barcoded libraries were constructed for each sample using the TruSeq RNA v2 kit (Illumina, San Diego, CA, USA). Libraries were size-selected for ~200 bp fragments. Sequencing was performed using Illumina NovaSeq to generate 150 bp pair-end reads. The sequencing depth was >25 million reads per sample for all samples.

### 2.4. Bioinformatics

NGS reads from the P65 male hippocampi in a prior study [26] were sequenced with 50 bp pair-end reads on a HiSeq2000 instrument. We hypothesized that changes in gene expression due to maternal ID and choline supplementation would have both sex-dependent and -independent responses. The sex-specific responses would contribute to differential outcomes of ID and choline supplementation between the sexes. Because the data were collected in two batches with different sequencing technologies, the sex-diet interaction effects might be confounded. To reduce the potential confounds, the following data preprocessing and analyses were performed: (1) Filtering and normalization—raw read counts were filtered to include only genes that were 300 bp or longer and expressed in at least one of the four treatment groups; (2) Statistical modeling—for differential gene expression tests, expression values were modeled by a generalized linear model (GLM) with a negative binomial distribution using either the dietary group as the variable if the model was built for each sex separately (16 RNA-sequencing (RNA-seq) libraries) or with all factors (iron, choline, sex, and their interactions) as variables using all 32 RNA-seq libraries. For the analysis of variance, a linear mixed model with the factors (sex, choline, iron, and their interactions) was used; (3) Analysis of differentially expressed (DE) genes—differential gene expression was tested using a quasi-likelihood F-test on the fitted GLMs and the following comparisons: Iron statuses ((ID, IDch) vs. (IS, ISch)) disregard of choline treatments for general iron effects, choline-supplementation ((ISch, IDch) vs. (IS, ID)) disregard iron statuses for general choline effects, iron by choline interaction, iron by sex interaction (sex-specific iron effects), and choline by sex interaction (sex-specific choline effects); DE genes were those with FDR < 0.05. These DE genes were used to perform unsupervised hierarchical clustering of the samples and to generate a heatmap depicting the counts per million (CPM) values of the DE genes in each group; (4) Analysis of variance—the contributions of each factor to gene variances were estimated using the R package variancePartition (VP). The top 2% genes (approximately 300 genes) with variances explained primarily by the analyzed variables (i.e., iron, choline, or sex) were selected for overrepresentation analysis (ORA) of GO terms; (5) Gene set enrichment analysis (GSEA)—all expressed genes were ranked using their differential expression test results by −log (*p*-value) * sign (fold change). Ranked gene lists were tested for enrichment against the Hallmark gene sets from MsigDB. An FDR < 0.25 cutoff was used for statistically significant set enrichment. A complete description of analyses and results is shown in Appendix A.

### 2.5. Ingenuity Pathway Analysis (IPA)

DE genes and ranked gene lists used in GSEA were mapped onto known biological processes using the “Core Analysis” feature of IPA, which includes >8.5 million studies (Qiagen, Redwood City, CA, USA). Ranked gene lists were also used to identify altered upstream regulators and known biomarkers in the hippocampus, cerebrospinal fluid (CSF), blood, and plasma/serum. Significant findings were determined using an absolute z-score ≥ 2.0 computed by Fisher’s exact test.

## 3. Results

### 3.1. Maternal ID and Choline Supplementation Altered Offspring’s Hippocampal Gene Expression in a Sex-Specific Manner

Effects of ID: NGS data from both sexes were combined to assess the long-term effects of fetal-neonatal ID on the transcriptional regulation in the adult hippocampus. After filtering and normalization, the combined dataset contained 15,060 loci. The combined dataset was analyzed by a GLM to test for DE genes between iron-deficient (ID, IDch, both sexes) vs. iron-sufficient (IS, ISch, both sexes) groups, which showed no gene with FDR < 0.05 (Figure 1A).

Effects of prenatal choline supplementation: To estimate the effect of choline, the combined dataset was tested for DE genes between choline-supplemented (ISch, IDch, and both sexes) vs. choline non-supplemented (IS, ID, and both sexes). One hundred and one genes (61 down- and 40 upregulated) showed significant changes in choline-supplemented groups (Figure 1A). Unsupervised clustering analysis of all samples using the identified DE genes showed a dendrogram with a non-uniform clustering of choline-treated groups (Figure 1B).

Sex-specific effects of choline supplementation: Analysis of sex and choline interaction showed a significant effect with 996 DE genes (Figure 1A). A violin plot of the top six genes with the lowest FDR values showed expected sex-differential effects (Figure 1C). A similar plot of 6 randomly selected genes showed no such interaction (Appendix A).

Assessing and managing confounding effects: Since male and female data were collected in different batches, we estimated the sex/batch contribution to variance and overall gene expression. Principle component analysis showed that sex/batch/technology can account for most of the variability in the combined dataset (Figure 1D). Analysis of variance showed that the variation across sex/batch/technology could account for a median of 14.6% of the variation in gene expression; whereas, the median effects of choline, iron, and their interactions were negligible (Figure 1E). To remove the potential confounding effects, we performed separate analyses of each sex.

### 3.2. Effects of Fetal-Neonatal ID and Choline Supplementation in Female Rats

NGS data from 16 female rats (4 from each treatment group) were analyzed. 14,965 of the 32,883 annotated genes passed the filtering process. Multidimensional reduction analysis showed a greater effect of choline than ID evidenced by a better group separation associated with choline than iron status (Figure 2A). Analysis of variance indicated that choline could account for 20% of the variability in the expression profiles of all female animals (Figure 2B). Testing for DE genes showed no significant ID effect (Figure 2C, (ID, IDch) vs. (IS, ISch)). Conversely, choline showed a clear effect on 1340 DE genes (629 down- and 711 upregulated) irrespective of their iron status (Figure 2C, (IDch, ISch) vs. (ID, IS)). Further assessment for the effect of choline on the IS group showed 153 down- and 243 upregulated genes (Figure 2C, ISch vs. IS). The unsupervised clustering heatmap showed expected patterns, where the expression of most DE genes followed the choline treatment (Figure 2D). Functional changes annotated by IPA indicate reduced anxiety and emotional behaviors, paired-pulse facilitation of synapse, and potential viability of neuroglia (Z-score = −1.13), accompanied by increased migration of cerebral cortical cells and motor function (Figure 2E).

To assess the rescue effects of choline, IDch females were compared to IS females; 243 (111 down- and 132 upregulated) DE genes were identified (Figure 2C, IDch vs. IS). An unsupervised clustering heatmap showed a clear separation between choline and non-choline groups (Figure 2F). To further identify the ID and choline interaction, libraries were compared between IDch and ID groups. 569 (312 down- and 257 upregulated) DE genes were found (Figure 1C, IDch vs. ID). Clustering using these DE genes with all libraries showed that samples were separated by choline supplementation (Figure 2G). Within the non-choline groups, samples were also separated depending on their iron status. In contrast, the choline-treated samples were not clustered following the iron status (Figure 2G). The findings indicate that choline normalized gene expression in the IDch group to that in the ISch group. The comparative analysis found 147 overlap-DE genes between IDch vs. IS and IDch vs. ID groups. These DE genes suggested activation of neuronal proliferation and migration (Figure 2H). IPA analysis of the non-overlap-DE genes indicates that the IDch group showed inhibition of neurodegeneration compared to the IS group and activation of cerebrovascular dysfunction but inhibition of dendritic growth compared to the ID group (Figure 2H).

### 3.3. Effects of Fetal-Neonatal ID and Choline Supplementation in Male Rats

NGS data from 16 male rats (four from each treatment group) were analyzed. 14,305 of the 32,883 annotated genes passed the filtering process. Clustering analysis by multidimensional scaling plot showed that the principal sources of variation among samples did not correspond with iron or choline status. Choline had a stronger effect on the expression profiles than iron status evidenced by a greater heterogeneity in the choline groups and less separation between the IS and ID samples (Figure 3A). Testing for DE genes showed no ID effect (Figure 3B, (ID, IDch) vs. (IS, ISch)). Comparison between choline vs. non-choline groups identified 132 DE genes (Figure 3B, (IDch, ISch) vs. (ID, IS)). These DE genes did not show the expected clustering pattern based on treatments (Figure 3C). Further comparative analysis of ISch vs. IS groups showed 37 DE genes that were clustered based on choline treatment, indicating an effect of choline (Figure 3D). The rescue effects of choline were analyzed by comparing IDch to IS or ID groups. These analyses found 54 (IDch vs. IS) and 276 (IDch vs. ID) DE genes (Figure 3B). Clustering analyses of DE genes in the IDch vs. IS comparison showed a clear separation between choline vs. non-choline groups (Figure 3E), but not in the IDch vs. ID comparison (data not shown). These findings indicate a larger effect of choline than ID on long-term gene expression.

### 3.4. Alternative Approach

The genome-wide variance among the samples from the combined RNA-seq dataset produced too few DE genes (FDR < 0.05). We utilized two additional approaches, including Overrepresentation Analysis and Gene Set Enrichment Analysis, to glean the altered biological functions attributable to sex, choline, iron, and their interactions.

### 3.5. Overrepresentation Analysis

The top 300 genes identified by VP analysis were tested for overrepresentation of GO terms. The joint dataset combining male and female rats showed a statistically significant overrepresentation of GO terms associated with synaptic membrane and membrane transporter complex in the group of genes with the highest variance explained by the choline-iron interaction (Figure 4A). These findings are driven primarily by the male data (Figure 4B) as the female data showed a lower number of enriched GO terms (Figure 4C). These findings suggested an interaction between choline and ID in adult rat hippocampus, which is more pronounced in males than females. Volcano plots showed that most, if not all, of these genes have a significant *p*-value (<0.05, Figure 4D–F). IPA analysis of the top 300 VP outlier genes from the joint dataset indicated that choline significantly activated the transcriptional regulator cAMP responsive element binding protein (CREB1) gene network, which was inhibited by ID-sex interaction (absolute z ≥ 2.0, Figure 4G). The glutamate ionotropic receptor NMDA type subunit 3A (GRIN3A) gene network was inhibited by ID-choline interaction. Among genes affected by choline-sex interaction, the inflammatory regulator transforming growth factor beta 1 (TGFb1) gene network was inhibited (Figure 4G). In males, ID activated the transcription factor fifth Ewing variant (FEV) gene network, which was mitigated by choline treatment (Figure 4H). In females, IPA analysis showed inhibition of alpha-synuclein (SNCA) gene network (z = −1.63) by choline supplementation (Ch), inhibition of intracellular cholesterol transporter Niemann-Pick C1 protein (NPC1) by ID-choline interaction, and activation of the G protein-coupled receptor adenosine A2a receptor (ADORA2A) gene network (z = 1.98) by ID (Figure 4I).

### 3.6. Gene Set Enrichment Analysis

*Effects of choline and ID*: Choline-driven gene expression changes showed more suppressed than activated gene networks (Figure 5A, Choline). Sex was also a factor for gene expression variation as evidenced by a reversal of specific biological activities (Figure 5A, Sex by choline). Specifically, gene networks associated with epithelial-mesenchymal transition, apical junction, and apical surface were downregulated in choline-supplemented females compared to males. On the other hand, gene networks associated with xenobiotic metabolism, fatty acid metabolism, and MYC targets V1 were upregulated in choline-supplemented females compared to males. ID-driven gene expression changes indicated more suppressed functions that were also influenced by sex. While the common trend for both sexes was a downregulation of pathways in ID (e.g., oxidative phosphorylation, DNA repair, and ROS pathway; Figure 5A, ID), the sex-by-ID interaction showed evidence of sex-specific effects (Figure 5A, Sex by ID). Particularly, MYC targets V1, oxidative phosphorylation, fatty acid metabolism, protein secretion, and interferon alpha response were upregulated in ID females when compared to ID males, indicating a sex and ID interaction. Volcano plots showed genes with a significant *p*-value (<0.05, Figure 5B).

*Sex-specific effects of prenatal choline and fetal-neonatal ID*: To confirm the sex-specific effects, choline- and ID-driven DE genes were analyzed separately by sex. In females, GSEA analysis showed that choline-induced changes in gene expression indicate more activated biological functions, whereas ID resulted in more suppressed functions (Figure 6A). Notable changes included activation of interferon-gamma (INFg) and tumor necrosis factor-alpha (TNFa) signaling in choline-supplemented groups, and activation of INFa but inhibition of TNFa in the ID groups (Figure 6A). In contrast to the effects on females, choline- and ID-driven gene expression changes showed fewer significantly altered biological functions in males, where there were more suppressed activities (e.g., protein secretion, and fatty acid metabolism) in the choline groups (ISch, IDch) but more stimulated activities (e.g., INFg, epithelial-mesenchymal transition) in the ID group (Figure 6B). In sum, results from the sex-separate analysis correlated with those from the joint GSEA, confirming the differential effects of choline and ID on specific pathways for female and male rats.

### 3.7. Biomarkers

Using the biomarker library available in the IPA database, ID- and choline-driven candidate genes identified by VP analysis (top 300 genes) were assessed for potential roles as markers of long-term effects (Table 1). In female rats, both ID, choline supplementation, and their interaction changed the expression of markers regulating inflammation (*Arg1*, *Cxcr4*, *Cx3cr1*, and *Pla2g7*). In male rats, altered expression of markers includes regulators of cellular growth and survival (*Anxa2*, *Eno2*, *Nrg1*) by ID, protein secretion (*Inhbb*) by choline (Ch); and synaptogenesis (*Pten*, *Pias1*) by ID-choline interaction.

## 4. Discussion and Conclusions

By assessing global changes in hippocampal gene expression, this study presents novel findings pertaining to long-term sex-specific hippocampal effects of early-life ID and prenatal choline supplementation on gene regulation and changes in pre-defined neurologic biomarkers. DE genes identified by the traditional approach with FDR value < 0.05 showed that major factors contributing to the long-term changes in hippocampal gene regulation were sex, choline supplementation, and iron status in that order. Interaction between sex and choline supplementation showed the most robust effects compared to the iron status by sex interaction or individual factors on hippocampal gene expression. Overall, female rats exhibited more effects of choline and ID than male rats. Because of the high variations among the combined samples due to integration from two separate RNA-seq batches using two different sequencing technologies, alternative approaches were applied to identify genes with variance driven by the analyzed variables (ID, choline, and their interaction) using genes deviated from the genome-wide trend [27], as well as to use ranked gene lists to identify altered gene networks by GSEA analysis. Both ID and choline supplementation showed similar patterns of sex-specific effects on changes in hippocampal gene networks and associated biological processes.

Little is known about the long-term effects of prenatal choline supplementation on the rat hippocampal transcriptome. Here we showed that prenatal choline-induced long-term sex-specific gene expression changes in adult hippocampal transcriptomes with greater effects in female than male rats. The robust effects in female rats suggest a more amenable epigenetic regulation, which indicates a more adaptable developing hippocampus in female than male rats. This adaptability may confer sex-specific advantages such as neurodevelopmental resilience in the advent of prenatal adverse exposures (e.g., nutritional deficiencies, maternal stress) [28,29]. In addition, prenatal choline produced more significant transcriptomic changes in adult rat hippocampus compared to fetal-neonatal ID in either sex, implicating an important role of epigenetic regulation in the backdrop of ID, given that choline is a known methyl donor that influences epigenetic modifications [30]. These findings could also provide a molecular explanation for the recovery of ID-induced impairments of neuronal structure and cognition even during the period of limited iron substrate [25,26,31].

Our study further expands the knowledge of the interaction between choline and iron status previously discovered [21,26,32] with a demonstration of sex-specific effects. In ISch females, choline might provide beneficial effects by altering the expression of genes that regulate emotions including anxiety and motor function. Choline could also reprogram anti-inflammatory responses with the activation of INFg and TNFa [33,34] in females (Figure 6). In this regard, there is evidence that prenatal choline supplementation mitigates fetal alcohol-exposure-induced pro-inflammatory responses in the adult rat hippocampus [35]. Thus, the role of prenatal choline in neuroinflammation deserves further analysis. On the other hand, choline could also produce undesired effects. In the ISch males, choline might induce adverse cellular effects similar to those observed in iron-deficient animals by suppressing the activity of oxidative phosphorylation and fatty acid metabolism [36,37,38] as well as activating cell adhesion and migration by increasing epithelial-mesenchymal transition. These findings indicate both benefits and potential risks of prenatal choline to the IS rats. Thus, the use of choline as a therapeutic agent needs additional investigation, particularly in the long-term gene expression changes that modulate neurobehavioral functions [26].

Regarding the rescue effects of prenatal choline on ID-altered gene expression, a greater number of genes was found when compared to ID than IS hippocampus (IDch vs. ID compared to IDch vs. IS) in both sexes, suggesting a robust rescue effect of choline on gene regulation, particularly among genes regulating MTOR and CREB signaling and inflammatory response. These effects of ID have been previously reported in models of perinatal ID [8,39,40]. Novel rescue effects of choline include the normalization of the ID-activated ADORA2 gene network in females and the FEV gene network in males. ADORA2 gene network is implicated in emotional behaviors and neurodegenerative disorders [41,42]. FEV gene network is implicated in depression and anxiety [43]. It is worthwhile to note the suppressing effect of choline on alpha-synuclein (SNCA), whose translation is regulated by the iron-response element (IRE) in the 5′-UTR of its transcript, and its overexpression is associated with various neuropathologies, such as Parkinson’s and Alzheimer’s diseases [44,45,46,47]. The finding suggests a potential mechanism for the neuroprotective effect of prenatal choline. Not all ID-induced effects (e.g., activated adipogenesis in females and suppressed oxidative phosphorylation in males) could be rescued by choline, suggesting the limitation of choline treatment.

Finally, a potentially important finding of the present study is the identification of biomarkers that are readily accessible in bodily fluids. Long-term specific changes due to ID (e.g., *Nrg1*), sex (e.g., *Cxcr4*, *Eno2*), or choline treatment (e.g., *Arg1*, *Pias1*) in the rat hippocampus indicate that these genes can be utilized as potential markers to gauge the health status of the hippocampus as well as other brain regions. Future studies can determine the relationship of these markers between blood and brain compartments following ID anemia. In this regard, our PIA model [8] would be an ideal system to determine the utility of these biomarkers as we could generate various degrees of ID anemia severity (i.e., 25 vs. 18% hematocrit). Determining the predictive values of these potential biomarkers from an easily accessible source (e.g., blood) would be critically important in the prevention and treatment of brain ID in order to reduce the life-long risks of poor neurobehavioral outcomes.

In conclusion, the power of the present study is in the global unbiased approach to identify differential gene expression changes due to ID, sex, and choline treatment. The long-term changes in gene expression, and their regulation, encompass various biological processes (e.g., synaptogenesis, inflammation). The present study has achieved our immediate goals in uncovering sex-specific effects of early-life ID and prenatal choline treatment on gene expression, laying down the groundwork by providing targets for future studies to address the causal relationship between these specific gene expression changes and hippocampal-dependent behavioral outcomes as well as more broadly functional consequences of the cortical-hippocampal-striatal circuitry.

## Figures and Tables

**Figure 1 nutrients-15-01316-f001:**
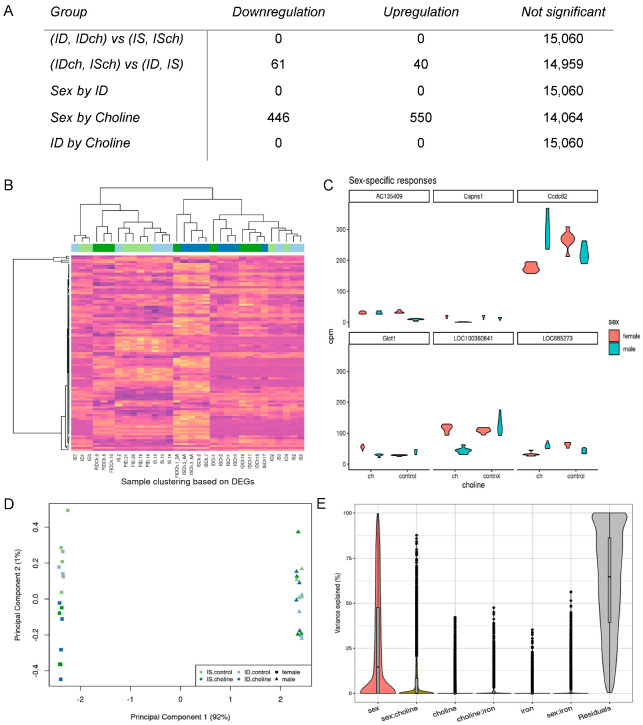
Bioinformatic integration of male and female RNA-seq datasets generated using different sequencing platforms. (**A**) Comparative analysis between groups showing a robust effect of sex-by-choline interaction on hippocampal gene expression. (**B**) Heatmap generated from unsupervised clustering showing a pattern that did not follow expected choline effects. Each bar represents gene expression in CPM, with purple being the lowest expression value and yellow being the highest. Blue and green bars at the top indicate the group assignment. (**C**) Top 6 representative genes showing sex-by-choline interaction effects. (**D**) Multidimensional plot showing a clear separation between sexes but not ID or choline treatment. (**E**) Analysis of variance revealed that variations in gene expression among combined libraries can be accounted for by sex.

**Figure 2 nutrients-15-01316-f002:**
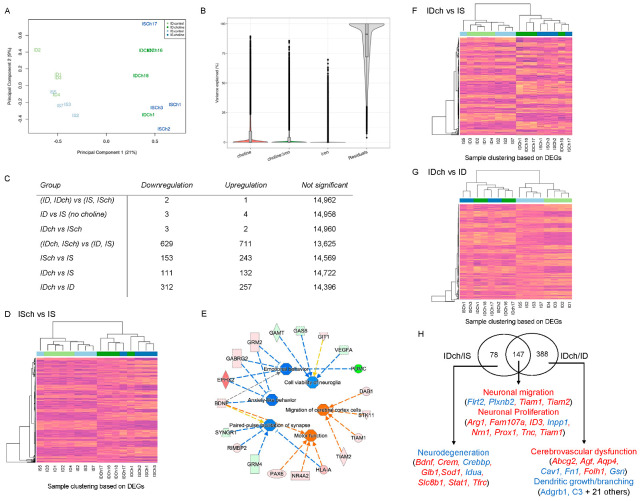
Effects of choline supplementation and ID on the female hippocampal transcriptome. (**A**) Multidimensional plot showing a clear separation between choline−supplemented and non-supplemented groups, but not with iron status. (**B**) Analysis of variance shows that choline and choline-by-iron interaction can account for changes in gene expression. (**C**) Table showing gene expression changes among comparison groups. (**D**) Heatmap of DE genes from ISch vs. IS comparison showing clear clustering between choline-supplemented and non-supplemented groups. (**E**) IPA annotated functional changes based on DE genes between ISch vs. IS comparison. Blue = Inhibition, Orange = Activation, Yellow = Finding inconsistent with the state of the downstream molecule, Grey = Finding not predicted, Solid and Dashed arrows = direct and indirect interactions, Green-to-Red = Downregulated-to-Upregulated genes. (**F**,**G**) Heatmaps showing distinct clustering of choline-treated samples using DE genes in IDch vs. IS or ID groups. (**H**) IPA analysis of overlapping DE genes between IDch vs. IS and IDch vs. ID groups showing the rescue effects of choline treatment on specific biological activities. In heatmaps, each bar represents gene expression in CPM, with purple being the lowest expression value and yellow being the highest. Blue and green bars at the top indicate the group assignment.

**Figure 3 nutrients-15-01316-f003:**
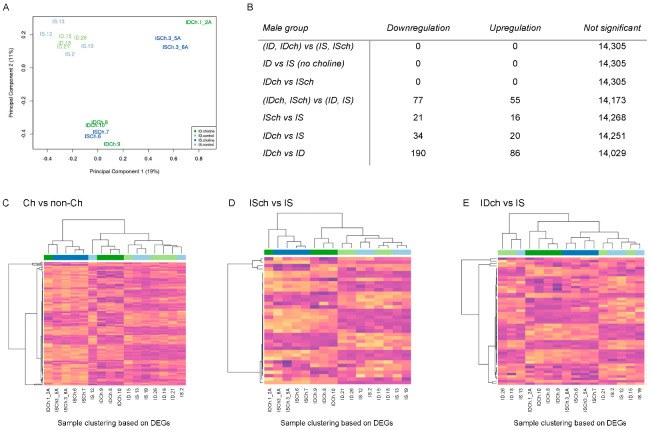
Effects of choline supplementation and ID on the male hippocampal transcriptome. (**A**) Multidimensional plot showing that the sources of variation among samples do not correspond with either iron status or choline treatment. Choline supplementation seems to have a stronger effect than an iron status on gene expression profiles as choline non−treated samples clustered together. (**B**) The table shows gene expression changes among comparison groups with the most affected in the IDch vs. ID comparison. (**C**–**E**) Heatmaps from DE genes among comparison groups show clear clustering among choline-treated IS samples but less clear in the ID samples. Each bar represents gene expression in CPM, with purple being the lowest expression value and yellow being the highest. Blue and green bars at the top indicate the group assignment.

**Figure 4 nutrients-15-01316-f004:**
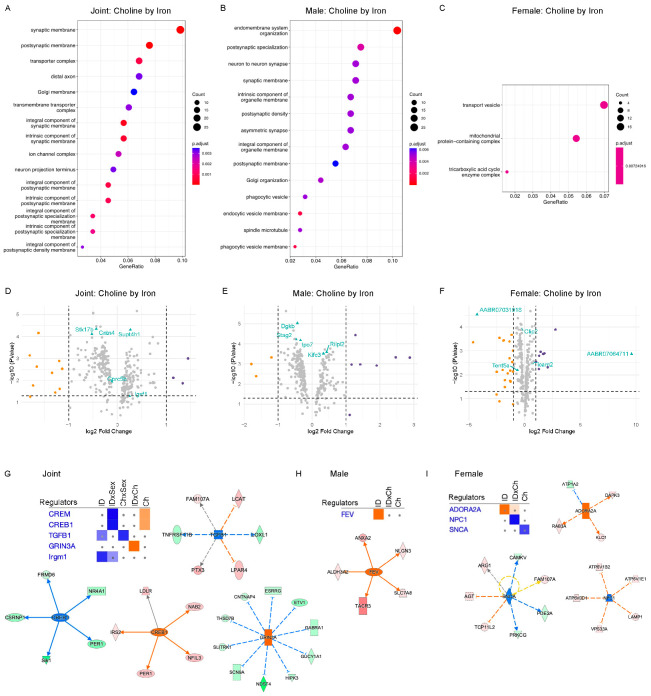
An alternative approach using top 300 VP outlier genes. (**A**–**C**) ORA analysis showing the effects of choline-by-iron interaction using sex-combined (**A**), male (**B**), and female (**C**) datasets. (**D**–**F**) Volcano plots showing genes with significant *p*-value (<0.05). (**G**–**I**) IPA-predicted changes of upstream regulators in the hippocampus using sex-combined (**G**), male (**H**), and female (**I**) datasets.

**Figure 5 nutrients-15-01316-f005:**
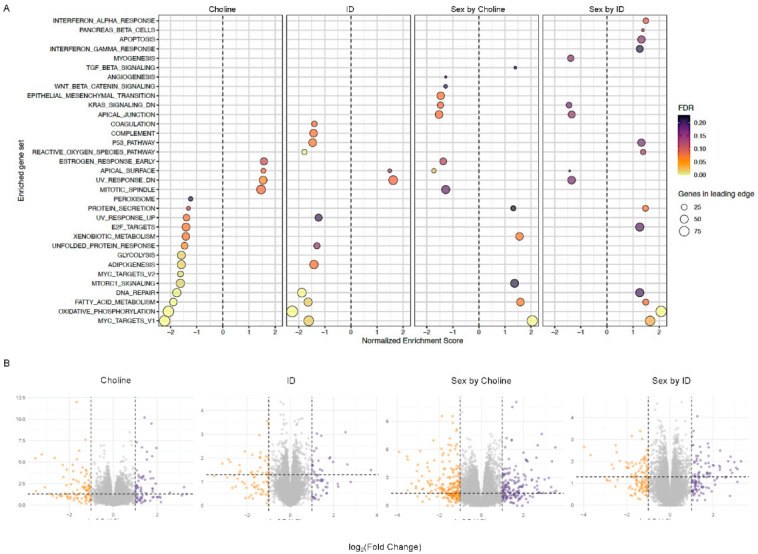
An alternative approach using ranked gene lists. (**A**) GSEA analysis using sex-combined datasets showing the effects of choline supplementation, ID, and their interaction with sex on enriched gene sets. Tests were made using males, IS, and non-choline-supplemented diets as a baseline. (**B**) Volcano plots showing genes with significant *p*-value (<0.05).

**Figure 6 nutrients-15-01316-f006:**
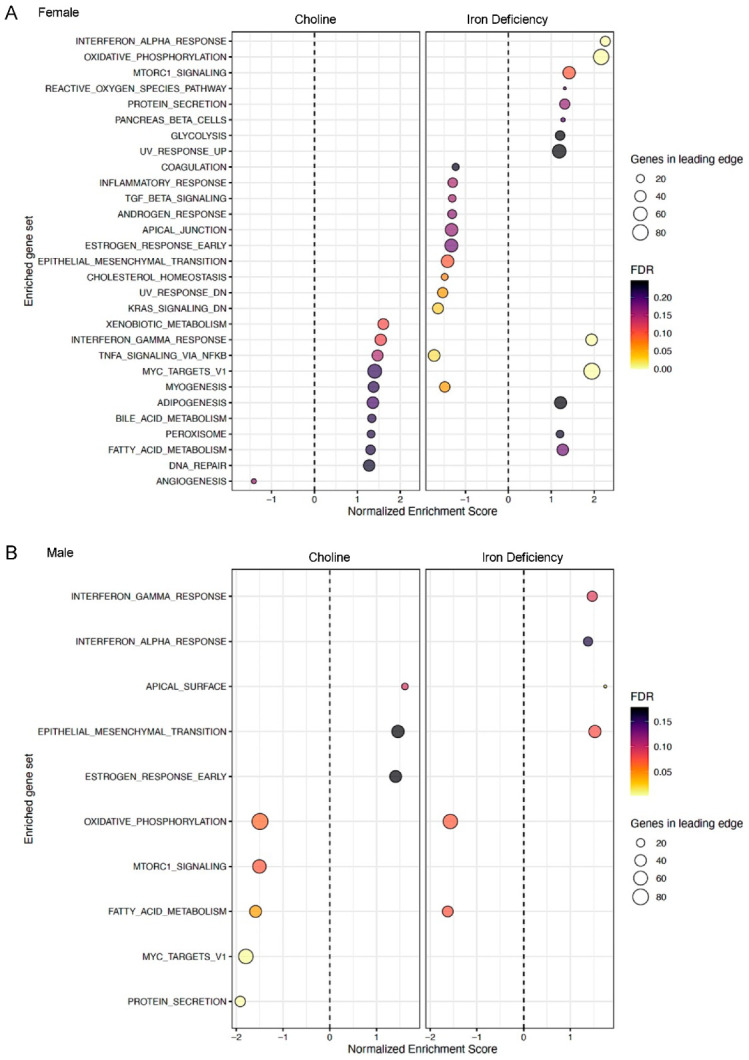
An alternative approach using ranked gene lists. GSEA plots showing the effects of choline supplementation and ID on enriched gene sets of female (**A**), and male (**B**) rats.

**Table 1 nutrients-15-01316-t001:** IPA-mapped known biomarkers found in humans, mice and rats.

Group	Gene Name	Symbol	Location	Family	Log_2_FC	*p*-Value	Tissues	Biomarker Application(s)
Female ID	C-X-C motif chemokine receptor 4	*Cxcr4*	Plasma Membrane	G-protein coupled receptor	0.61	0.006	B, Hc	diagnosis
Female Ch	Angiotensinogen	*Agt*	Extracellular Space	Growth factor	0.50	0.000	B, P/S, Hc	efficacy
	Arginase 1	*Arg1*	Cytoplasm	Enzyme	1.15	0.000	B, P/S, Hc	unspecified
	C-X3-C motif chemokine receptor 1	*Cx3cr1*	Plasma Membrane	G-protein coupled receptor	−0.55	0.000	B, P/S, Hc	unspecified
	GLI family zinc finger 2	*Gli2*	Nucleus	Transcription regulator	0.98	0.000	B, Hc	efficacy
	Integrin subunit beta 4	*Itgb4*	Plasma Membrane	Transmembrane receptor	0.99	0.000	B, Hc	diagnosis
	Prospero homeobox 1	*Prox1*	Nucleus	Transcription regulator	0.78	0.000	B, Hc	diagnosis, disease progression
Female IDxCh	Early growth response 1	*Egr1*	Nucleus	Transcription regulator	−0.75	0.000	B, P/S, Hc	diagnosis, efficacy
	Gelsolin	*Gsn*	Extracellular Space	Other	−0.59	0.000	B, CSF, P/S, Hc	disease progression, efficacy
	Phospholipase A2 group VII	*Pla2g7*	Extracellular Space	Enzyme	0.61	0.002	B, Hc	diagnosis, efficacy
Male ID	Annexin A2	*Anxa2*	Plasma Membrane	Other	0.73	0.006	B, P/S, Hc	diagnosis
	Enolase 2	*Eno2*	Cytoplasm	Enzyme	−0.66	0.000	B, P/S, Hc	diagnosis, efficacy, prognosis
	Gelsolin	*Gsn*	Extracellular Space	Other	−0.51	0.000	B, CSF, P/S, Hc	disease progression, efficacy
	Neuregulin 1	*Nrg1*	Plasma Membrane	Growth factor	1.06	0.002	B, Hc	diagnosis, response to therapy
Male Ch	Inhibin subunit beta B	*Inhbb*	Extracellular Space	Growth factor	0.68	0.002	B, P/S, Hc	efficacy
	MAF bzip transcription factor	*Maf*	Nucleus	Transcription regulator	0.56	0.005	B, Hc	unspecified
Male IDxCh	Endothelin 1	*Edn1*	Extracellular Space	Cytokine	0.90	0.008	B, P/S, Hc	diagnosis, efficacy, prognosis
	Protein inhibitor of activated STAT 1	*Pias1*	Nucleus	Transcription regulator	−0.58	0.001	B, Hc	unspecified
	Phosphatase and tensin homolog	*Pten*	Cytoplasm	Phosphatase	−0.52	0.000	B, Hc	diagnosis, disease progression, efficacy, prognosis, response to therapy
	Sex hormone binding globulin	*Shbg*	Extracellular Space	Other	0.51	0.009	B, P/S, Hc	efficacy, safety

Selection criteria include absolute log_2_FC ≥ 0.50, *p* < 0.05. Abbreviations: Blood (B), Cerebrospinal fluid (CSF), Plasma/Serum (P/S), and Hippocampus (Hc).

## Data Availability

NGS data will be deposited to NCBI following publication.

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
