# Peer review of "Sex-Specific Effects of Early-Life Iron Deficiency and Prenatal Choline Treatment on Adult Rat Hippocampal Transcriptome"

_nutrients, 2023, doi:10.3390/nu15061316_

Round 1

Reviewer 1 Report

Liu et al. analyzed and compared long-term changes in hippocampal gene regulation between adult male and female rats that were iron-deficient versus iron-sufficient during fetal-neonatal period and between those with and without prenatal choline treatment. This review article is well written and carries novel findings.

Only a few typos are detected:

What are the birth weights and sex ratio of the pups? The authors did not report this finding in their previous studies.

What are the serum iron level and related parameters, such as TIBC? Reduced brain iron concentration in P10 pups and normal hematologic status by P28 were reported in their previous study. More iron-related data is helpful in the clinical context.

Reviewer 2 Report

The authors have carried out a good piece of research on the relationship between sex, prenatal iron deficiency and choline treatment on adult tat hippocampal transcriptome. However, in order to publish the work, the following considerations should be taken into account.

The researchers should clearly explain why they have changed the sequencing platforms in the case of female rats. Why did they not use the same one as in the case of male rats? Much of the article focuses on justifying that the results obtained are comparable; this would not have been necessary if the same methodology had been used in both cases (male and female).

The number of animals used in the study is unclear: what is the initial number of dams, why were only 16 individuals of each sex used to carry out the study?

In Figure 3 the 16 male individuals included in the study are identified; what does FID and FIDch stand for, could the presence of the letter "F" indicate female, have females been included in the comparison by mistake or is it simply a typing error?

The figures need to be improved, the colours used are confusing and Figure 2 is illegible.

It represents

I hope my comments have helped to improve the paper.

kind regards
